# Network percolation reveals adaptive bridges of the mobility network response to COVID-19

**Hengfang Deng[1], Jing Du[2], Jianxi Gao[3]\*, Qi Wang[1]\***

**1** Department of Civil and Environmental Engineering, Northeastern University, Boston, MA, United States of America, **2** Department of Civil and Coastal Engineering, University of Florida, Gainsville, FL, United States of America, **3** Department of Computer Science and Center for Network Science and Technology, Rensselaer Polytechnic Institute, Troy, NY, United States of America

\* gaoj8@rpi.edu (JG); q.wang@northeastern.edu (QW)

**Data Availability Statement:** The data is provided by the third party, Cuebiq Inc. All data were collected through a CCPA and GDPR compliant framework and utilized for research purposes. Our usage agreement with Cuebiq does not allow us to

## Abstract

Human mobility is crucial to understand the transmission pattern of COVID-19 on spatially embedded geographic networks. This pattern seems unpredictable, and the propagation appears unstoppable, resulting in over 350,000 death tolls in the U.S. by the end of 2020. Here, we create the spatiotemporal inter-county mobility network using 10 TB (Terabytes) trajectory data of 30 million smart devices in the U.S. in the first six months of 2020. We investigate the bond percolation process by removing the weakly connected edges. As we increase the threshold, the mobility network nodes become less interconnected and thus experience surprisingly abrupt phase transitions. Despite the complex behaviors of the mobility network, we devised a novel approach to identify a small, manageable set of recurrent critical bridges, connecting the giant component and the second-largest component. These adaptive links, located across the United States, played a key role as valves connecting components in divisions and regions during the pandemic. Beyond, our numerical results unveil that network characteristics determine the critical thresholds and the bridge locations. The findings provide new insights into managing and controlling the connectivity of mobility networks during unprecedented disruptions. The work can also potentially offer practical future infectious diseases both globally and locally.

## Introduction

The ongoing pandemic continues to wreak havoc across the globe, and the U.S. has suffered the highest impact among all countries with over 20 million infections 350,000 deaths [1, 2]. Despite the sustainable efforts in forecasting and containing the deadly virus, it has been challenging to predict COVID-19 spread. The epicenters and "hot spots" have shifted from Seattle to NYC, to the southern parts of the country in a short few months [3, 4]. The unprecedented and unforeseeable changes have caused significant challenges to stop the deadly virus and contain its impact on the U.S. economy and society.

Historically, a large and diverse literature has examined the relationship between human mobility and the diffusion of infectious disease, including SARS, seasonal influenza, and

make public or otherwise share the anonymized mobile phone data used in this study. Researchers interested in aggregated data and/or summary statistics, where permitted under said agreement, should contact the corresponding authors or Cuebiq inc. at https://go.cuebiq.com/request-demo. The authors of the present study had no special access privileges in accessing these datasets which other interested researchers would not have.

**Funding:** Q.W. acknowledges the support from National Science Foundation (No. 2027744). https://www.nsf.gov/awardsearch/showAward? AWD_ID=2027744&HistoricalAwards=false. J.D. acknowledges the support from National Science Foundation (No. 2027708). https://www.nsf.gov/awardsearch/showAward?AWD_ID=2027708&HistoricalAwards=false. The funders had no role in study design, data collection and analysis, decision to publish, or preparation of the manuscript.

**Competing interests:** The authors have declared that no competing interests exist.

malaria [5–10]. The prior successes have inspired a number of studies examining how individual travel behaviors contribute to the spread of COVID-19 in the U.S., China, Italy, and other countries [11–17]. These studies demonstrate that human mobility, to certain extent, can forecast COVID-19's transmission trends [4, 11, 12, 14, 18]. Thus, the understanding, modeling, quantifying, and control of human mobility combined can potentially reduce the transmission of COVID-19 [19–24].

Despite the importance of establishing a quantitative foundation for understanding human mobility patterns, predicting it still faces at least two challenges. Firstly, human mobility is a complex behavior with multi-scale dynamics. Although demonstrating some universal patterns [25–32], human mobility varies significantly on the regional and community levels owing to sociodemographic factors [33–35], economic inequality [33, 36, 37], geographical settings [35], social interactions [38, 39], the availability of transportation infrastructures and mobility options [40]. The complexity is particularly critical for COVID-19 has unprecedentedly changed human mobility. Social distancing and travel restrictions have changed travel behaviors and might cause co-evolution between the mobility dynamics and COVID-19 spread [12, 41]. Secondly, data availability has constrained human mobility research. There is a shortage of large-scale and longitudinal data sets with rich samples [19] for predicting the spread of infectious diseases. The data is especially critical to track and capture the significant and dynamical disruptions of human mobility during COVID-19 in the U.S.

In the realm of network research, percolation theories are critical for identifying hierarchical structures and determining any discontinuous phase transition inside complex systems. There have been various studies using the percolation process in modeling real-world phenomena such as traffic and routing networks [42, 43], the spread of infectious disease [44, 45] and cascading dynamics [46, 47]. The bond percolation process has been commonly used in the epidemic modeling field entails deleting links at various cut-off values in order to find the critical threshold at which the network undergoes a structural shift such as break down of the large network into smaller components [48]. Since the presence of a link implies infection, such a threshold is closely tied to the emergence and disappearance of giant disease clusters. In the context of epidemic modeling, utilization of percolation theory combined with real-world mobility data can facilitate understanding the hierarchy of dynamic mobility networks and thus unearth the structure transition patterns [45, 49].

We analyze anonymous mobility data from de-identified and opted-in smartphones across the continental states' area to estimate mobility networks. We thereby capture mobility dynamics and assess the levels of changes in different regions during the first six months of 2020. We study the connectivity of daily mobility networks using percolation theory [50–52] originated from statistical physics [42, 53–55]. Surprisingly, we observe abrupt and critical phase transitions marked by large clusters disconnected from the giant components (*G.C.s*) of mobility networks. Our analyses mainly focus on the topology of the mobility network to find adaptive bridges that can lead to abrupt collapses of mobility networks. Our numerical method enables us to uncover fundamental patterns of the complex mobility networks under the influence of significant perturbations and potentially provide novel strategies for managing human mobility on the national level. Which could help to prevent further the spread and possible resurge of COVID-19 and future infectious diseases.

## Results

### Mobility data

We utilize six months of anonymized and privacy-enhanced human mobility data, from January 1 to June 30, 2020, in the continental U.S. to construct the daily flux network on the county

level. The anonymized, de-identified data set provided by Cuebiq reports location in real-time and is crowd-sourced from over 30 million devices opted-in to anonymous data sharing for research purposes through a CCPA and GDPR compliant framework. We also record the total detected devices within each county and compare the numbers with 2018 American Community Survey (ACS) data for validation. On average, we witnessed over 15 million devices per day. The Pearson correlation coefficient between the county population and the users is approximately 0.95 from March through June, shown in S1 and S2 Figs in S1 File.

We find the median of the locations from each user's daily movement trajectory (i.e., the location of the medians of longitude and latitude) and compute the primarily located county on each day (see Fig 1a). Thus, we establish a link between user $U_i$ and county $C_j$ if the user's median location is in county $C_j$. We then build a county-user bipartite network of all users and counties for each day, as shown in Fig 1b. Finally, we map the inter-county mobility network of the day $k$ based on the county-user bipartite networks of the day $k − 1$ and day $k$. We create a directed link from county $i$ to county $j$ if at least one user (traveler) was in county $i$ on day $k − 1$ and in county $j$ on day $k$. The weight $W_{ij}$ is the number of travelers from county $i$ to county $j$, as shown in Fig 1c. For example, both travelers $U_0$ and $U_3$ were in county $C_1$ on day 0 and in county $C_2$ on day 1, so $W_{12} = 2$ and the connecting travelers are $U_0$ and $U_3$ in Fig 1c. We also obtain the corresponding undirected network for each day (Fig 1d) with weight as the sum of the numbers of travelers in both directions. In this work, we are interested in the strongly connected component (*SCC*) of directed networks and giant component of the undirected network. To eliminate random fluctuations, we average the networks from 7 consecutive days (see S3 and S4 Figs in S1 File for details).

## The spatiotemporal patterns of mobility network in the U.S

These daily inter-county mobility networks allow us to capture the day-to-day dynamics of mobility patterns. Some changes in the networks reflect the influence of the breakout of COVID-19 in the United States. Fig 2a shows the network structure in the week of Feb. 10, and Fig 2b one in the week of Mar. 30. The connectivity, indicated by edge weight $W_{ij}$, plummeted to a record low, with the second network exhibiting a much higher sparsity level. Although we observe that the edge weights follow truncated power-law distributions both before and after the national emergency declaration (Fig 2c), the distributions of travels have changed significantly. Travels below 50 km have increased by almost 400% unweighted (Fig 2d) or 96.1% weighted by the numbers of travelers (Fig 2e) while long-distance travels (i.e., >1000 km) dropped to 39.2% unweighted (Fig 2d) or 35.2% weighted by the numbers of travelers (Fig 2e). Overall, these results indicate that the fraction of inter-county trips decreases while intra-county traffic increases significantly as the stay-home orders become effective.

The changes are quantified in two network metrics: the total influx at each node and the sum of edge weights (i.e., the total number of travelers). To mitigate the variation by the days of a week, we construct the baselines using the data from the first two months and compute the average values for each day of the week. The changes are then calculated as the daily shift (in percentage) against the baselines. Fig 2f–2h shows the normalized perturbation on human travel behavior. However, the overall number of active devices experienced a small decrease (less than 5%, see Fig 2f), the in-degree of the nodes experienced a 20% decrease after the declaration of national emergency on March 13, 2020 (Fig 2g). The dive continued until the end of March and early April, reaching a record of a 50% decline before recovering. The daily number of inter-county travelers (i.e., total influx) also followed a similarly steep drop after mid-March and reduced by approximately 60% (Fig 2h). While the drop lasted about two weeks, the mobility recovery took about two months. We observe a steady increase from April

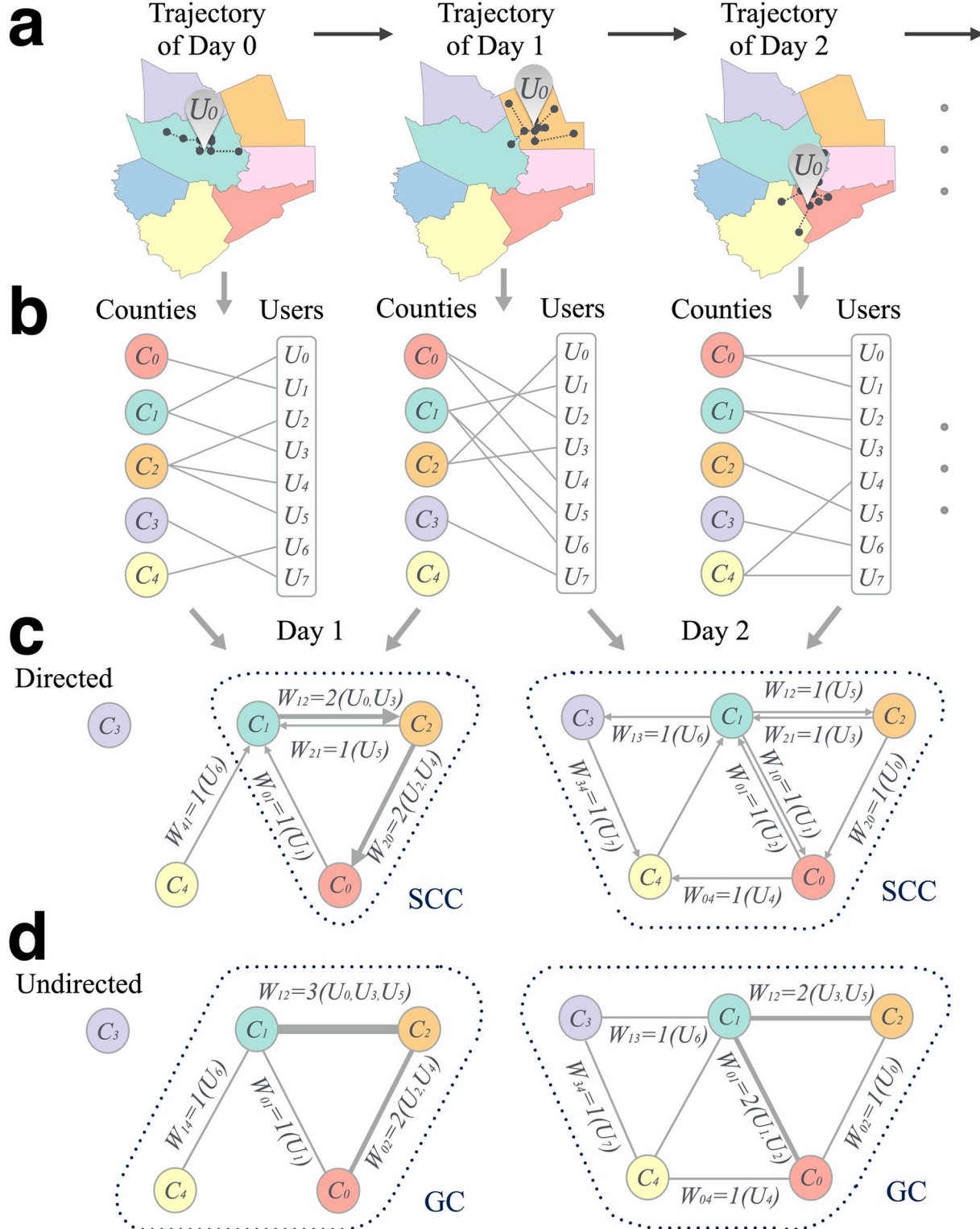

**Fig 1. Mobility network. a**, one user's trajectories in two consecutive days. We compute the median of the longitudinal records as the prime location and geotag to the corresponding county. **b**, we iterate such process for all users and construct the bipartite graphs based on collective mobility. **c**, by comparison of two consecutive days' bipartite graphs, we aggregate the inter-county travel information to a weighted and directed graph. The weights are the numbers of inter-county travelers. **d**, the undirected graph with weight as the sum between each pair of counties.

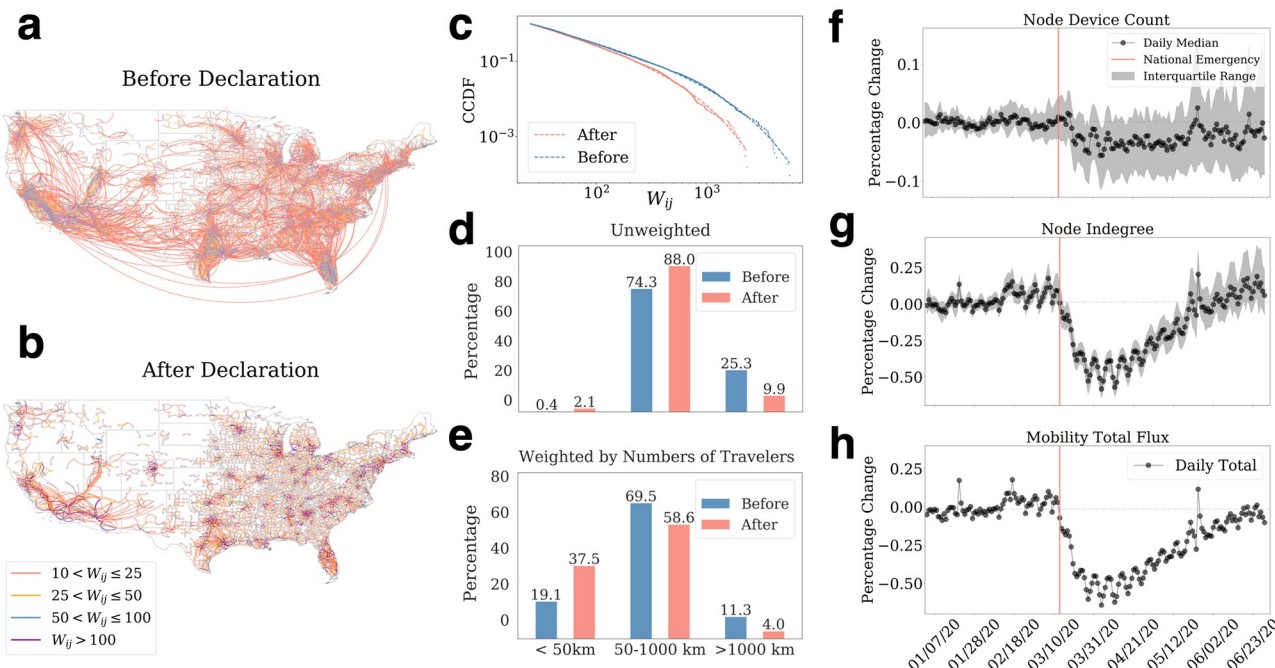

**Fig 2. Inter-county mobility patterns reflect social distancing and travel restrictions. a**, weekly average flux from Feb. 10, to Feb. 17, 2020. **b**, weekly average directed graph from Mar. 30, to Apr. 6, 2020. The mobility connections weakened after the National Emergency Declaration that took place on Mar. 13, 2020. Also, both long-distance and short-distance travel (i.e., links) have reduced. **c**, the distributions of edge weights for the two weeks. Both distributions follow the truncated power-law distributions with $\alpha = 1.93$ for the one before and 1.75 for after, indicating that people's movements have been disrupted significantly even though the fundamental patterns remained the same. **d**, the spatial distances of the unweighted links of the two weeks. The proportion of short-distance links increased from 0.4% to 2.1% while the middle range link's proportion rose from 74.4% to 88.0%. In contrast, the proportion of edges crossing long distances dropped significantly from 25.3% to 9.9%. **e**, spatial distances of the weighted links of the two weeks. The proportion of short-distance travel increases significantly to 37.5% after the social distancing rules. The fractions of mid and long-distance travel experienced about a 9.9% and 7.3% decrease, respectively. **f-h**, temporal changes of the inter-county mobility network. They are the daily percentage change of the total number of devices in each node (i.e., county) (**f**), node in-degree (**g**), and total inter-county mobility flux (**h**). Red lines are the day of the national emergency declaration. While the data only lost a small number of devices, the two measures both plummeted more than 50% within two weeks before recovering.

to the end of June as different regions in the U.S. started reopening. Inter-county mobility measures have almost returned to their levels before the pandemic by the end of June (Fig 2h). In summary, the metrics' changes reflect that the mobility network experienced an abrupt perturbation after the national emergency declaration before recovering.

## Connectivity of daily mobility networks

The observed perturbation in mobility networks largely aligns with our intuitive expectations: the pandemic has caused forceful and yet relatively short-term changes in indegrees and total flux. However, we still do not know whether *the disruption has caused phase transition in the mobility network or not*. To answer these questions, we examine the connectivity mobility networks by employing percolation theory. We remove the links in our inter-county mobility network if their weights are less than a given threshold $q$. When $q = 0$, all the counties are in the same *SCC*, and the *SCC* decreases as we increase the threshold $q$. In Fig 3a, we take the aggregated network from Feb. 10 to Feb. 17, 2020, as an example. Surprisingly, we observe an abrupt phase transition when $q$ is greater than a critical threshold $q_c$. A large cluster of over 600 counties is disconnected from the largest *SCC*, as shown in Fig 3a. Consequently, the size of the second strongly connected component (*SSCC*) suddenly reaches a peak at this critical point (red

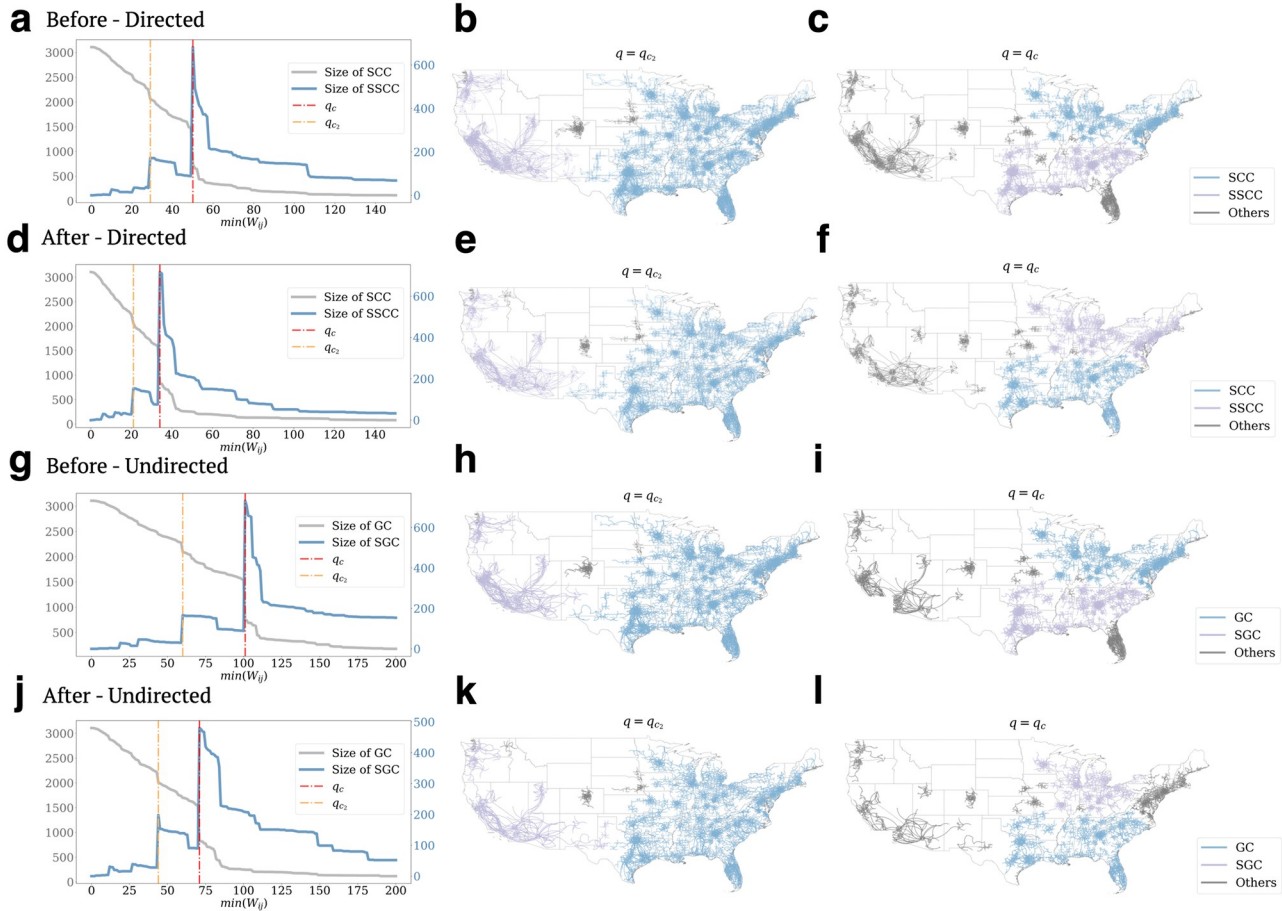

**Fig 3. Percolation of mobility networks. a** shows the sizes of the largest two components with the change of $q$ in the week of Feb. 10 to Feb. 17, 2020. There are two critical thresholds: $q_c$ at which *SSCC* experienced the largest size increase and $q_{c2}$ at which the size of *SSCC* reached the second largest peak. The networks at two thresholds are shown in **b** and **c** respectively. We show the top three sub-components, size-wise, which are in blue (largest), purple (second-largest) and grey (fourth-largest) at both thresholds if they exist. **d**, **e** and **f** are the percolation directed networks in the week of Mar. 30 to Apr. 6, 2020. **g-i** demonstrate similar percolation patterns for the same week of data before National Emergency using undirected graphs where **g** highlights the similar percolation phenomenon; The undirected graph results after the mobility perturbation are shown in **j-l** and we can observe that across all scenarios at $q_{c2}$, a large sub-component on the west coast states detached from the network while at $q_c$ at least three major sub-components are separated. These large clusters are similar despite the decrease of the value of $q_c$ after Mid March.

line in Fig 3a). The abrupt changes echo the disconnection of mobility between the Midwest and the South regions. Since $q_c$ measures the minimum flux needed to maintain the connection with the *SCC* of the mobility network, the clusters represent geographical regions consisting of counties with substantial traveler flux (Fig 3b). In the percolation curve, we also observe another notable size jump of *SCC* regularly occurs when the threshold is smaller than $q_c$, denoted as $q_{c2}$ (orange line in Fig 3a). At $q = q_{c2}$, both sizes of the *SCC* and *SSCC* experience significant changes, coinciding with the division of human mobility between the West and other regions (Fig 3c). We consistently observe similar percolation curves in the corresponding undirected networks; Fig 3g–3i illustrate a similar hierarchy of sub-components with different numerical scales of the critical thresholds.

The results also reveal the critical transition in mobility networks under the influence of COVID-19. In Fig 3d–3f and 3j–3l, we show the percolation of the directed network and corresponding undirected, respectively, in the week of Mar. 23, 2020. Comparing to the values in

the week of February, $q_c$ dropped from 53 to 30, and $q_{c2}$ decreased from 30 to 18. The significant decreases indicate that the mobility networks experienced strong perturbations; substantial populations have limited their travels within their home counties to reduce their exposure risks of COVID-19. Even though the geographical distributions of the clusters appear to be similar (Fig 3e and 3f) with the patterns in February, they are divided into sub-components at a lower threshold. By comparing the networks in Fig 3**e**, 3**f**, 3**k** and 3**l**, we observe analogous community structures despite perturbations. The novel abrupt phase transition discovered in the mobility network indicates that a small increase of flux threshold (i.e., mobility between counties) can lead to catastrophic collapses in network connectivity. Furthermore, this novel percolation is universal in the U.S. mobility network each day, and we show the results of some other days in S3–S6 Figs in the S1 File. Thus, the critical transition needs to be monitored more closely during crises since the mobility network becomes vulnerable during COVID-19 as the critical transition can happen at a significantly lower threshold of $q_c$. It, therefore, raises the critical questions: *what the critical points are and where the critical links are located in the network.*

### Temporal progression of percolation metrics

Next, we consider the effect of temporal property and explore the adaptation of the mobility network. Fig 4 shows such adaption. At the beginning of 2020, human mobility remained stable, and the value of $q_c$ stayed around 53 for directed and 107 for undirected graphs until Mar. 8, 2020. In this period, the median edge weights of the *GC* also remain the highest around 217 and 108 for *SSCC* (Fig 4b). The size of the largest component was around 636 (Fig 4c).

In the three weeks after Mar. 8, 2020, COVID-19 forced the U.S. population to practice social distancing and spend more time at home. The change is evident in Fig 4a as $q_c$ started to decrease drastically following the national emergency declaration and the stay-at-home orders from several states and many local jurisdictions. The critical threshold reached its lowest point, 30 for directed and 53 for undirected networks by the end of March. Since $q_c$ measures the minimum threshold to reach percolation criticality, the mobility network became significantly more vulnerable. The median values of the largest component weights dropped to 66 for directed networks and 112 for undirected ones, both shedding approximately 50%. We observe a similar pattern of decreases in the median edge weights (Fig 4b). The drop coincides with our observation in Fig 2g and 2h. It is worth noting that the largest component size appears to show opposite trends and fluctuate more than the other two variables despite experiencing similar stages of changes. The sizes of the largest connected components remained high for about two weeks, even after the number of domestic travelers started climbing on Mar. 31, 2020. During this time, the mobility is steadily increasing, while sub-components from the previously largest component are prone to merge. We observe that both the *GC* and *SSCC* sizes at $q_c$ increased from 636 to above 1,000 after the national emergency declaration (Fig 4**c**). This increase is likely due to the reduction in critical threshold; many localized sub-components are more prone to be integrated into the major clusters even with limited travels.

During the entire month of April, the mobility network stayed stable in a new phase after two weeks of drastic changes. The value of $q_c$ stayed low and steady for about three weeks (Fig 4**a**) following a fast drop. This pattern is observed for the median edge weights (Fig 4**b**) as well. The size of *GC* remained high yet also stable during this time (Fig 4**c**). Noticeably, the recovery delays are different from the basic mobility metrics measured in Fig 2**g** and 2**h**. Both mobility metrics immediately started to recover continuously after hitting the bottoms at the end of March. In contrast, the percolation features suggest that the overall network stayed at the low point for three more weeks before first experiencing a jump. The discrepancy again suggests

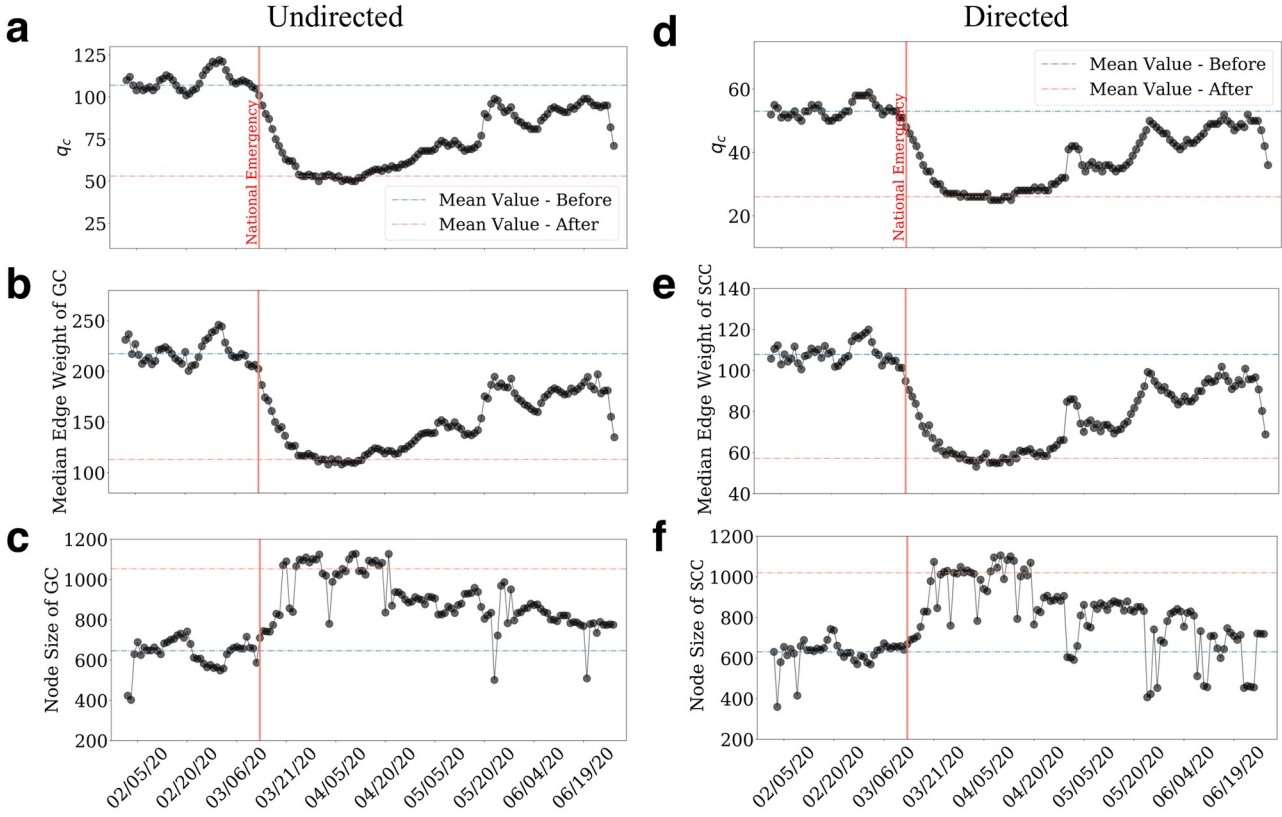

**Fig 4. The changes of the percolation metrics using a 7-day moving window to smooth out the weekend/weekday fluctuations. a** and **b**, the changes of $q_c$ with time and the median edge weights of the largest component which reflects the connection strength. **c**, the changes in the sizes of the largest strongly connected component for undirected graphs. **d,e** and **f** show the directed graph's scenario for $q_c$, median edge weight and largest component node size. Grey lines indicate the time series of the undirected graphs while blue ones are for directed graphs. The red vertical line highlights the time of the national emergency declaration (March 1, 2020) while the blue and red horizontal lines indicate the mean value of each feature before and after the declaration.

that the recovery of the percolation criticality of human mobility follows a phase transition process rather than continuous change. The phase transition offers a unique and crucial perspective on how to manage people's movement during a pandemic. Despite the fact that mobility changes on a continual basis, the complex networks produced by human mobility have been found to undergo sudden phase transitions. This suggests that a subtle shift in mobility could lead to wide-scale reconnections of human mobility networks, resulting in the spread and resurgence of COVID-19 across broad geographies.

By the end of April, all three metrics started to recover (Fig 4a–4f). Both $q_c$ and median edge weight recovered to about 85% of the original state by the end of June. Such recovery in mobility concerns as large parts of the country are still in their early stages of reopening, highlighting the challenges of curbing overall mobility in the long run. The resumption of human mobility might also contribute to the resurge of COVID-19 cases in the country.

## Recurrent critical bridges and their adaptations

The results so far demonstrate that the inter-county mobility network experienced abrupt phase transitions and the thresholds almost returned to their pre-COVID level by the end of June. Unlike the percolation of classic random or spatial networks, the percolation of inter-

county mobility network exhibit a discontinuous, abrupt phase transition. In a continuous change, a small fraction of link removal only leads to a slight size decrease of *SCC*. In contrast, in an abrupt phase transition, a small fraction of link removal may cause a catastrophic collapse of the network at the critical point. Due to the unpredictable structural changes, it is crucial to identify these vital links and pinpoint their removals in the temporal network. We propose a new method of identifying *recurrent critical bridges* that are critical links connecting the mobility network clusters. These links function as corridors for mobility across different geographies. Understanding their changes sheds new light on mobility perturbation under external perturbations. Also, managing and controlling the flows of a few critical bridges can be effective measures in containing COVID-19 spread and future infectious diseases.

Unlike its counterparts in static networks, finding the critical links in dynamic mobility networks is challenging. The network's structure was affected by COVID-19 spread and local policies, and the perturbation evolved both temporally and spatially. We address this issue by considering the network stricture at percolation criticality $q_c$, which is the backbone of the original network [53]. We define the critical links as the edges connecting the largest and second-largest clusters when the values of $q$ are just below $q_{c2}$ and $q_c$. These edges would be removed once $q$ reaches the thresholds. This method sometimes identifies links that only appear occasionally owing to the inherent randomness of human mobility. The frequencies of recurrences can be found in S7 Fig in the S1 File. Therefore, we use the frequency (i.e., the recurrence rate) of each identified link to determine the overall significance of the links; a link is only critical if it is identified for more than 10% of the days of the study period. Given the concurrence of most links of both undirected and directed graphs, we report only the undirected links. Eight recurrent connections are considered critical bridges (see S8 Fig, S1 and S2 Tables in S1 File). The recurrent critical bridges suggest that only a limited number of potential bridges emerge between the sub-graphs despite the dynamical percolation nature of human mobility.

We find that such recurrent critical links have adapted to the disease outbreak of COVID-19. The emergence of new links is usually caused by the discrepancy of mobility patterns between two sub-components. The surge of COVID-19 cases in one region often caused more restriction and perturbation in its population's mobility. The difference in mobility responses between two subunits is likely to result in unbalanced changes and the emergence of a new critical link. Fig 5a shows that prior to the COVID-19 outbreak (i.e., Stage 0), the U.S. mobility network was segmented into three large components: (1) the West particularly centered around the Pacific region, (2) the Midwest region, and (3) a large part of Eastern and Southern regions. We also observe that three of the bridging links were located at or close to the borderline of the components rather than the center. The only exception is the link connecting Maricopa County, Arizona, where the city of Phoenix is located, and Doña Ana County, New Mexico.

In Stage 1, the critical bridges largely stayed the same, except that Georgia became less critical (Fig 5b). In this stage, The connectivity dropped almost universally across the country after the national emergency declaration. The universal change reduced the connectivity within and among the identified components in the network and yet did not change the topology and the links.

In Stage 2, the critical bridges started to evolve due to the imbalanced perturbation in mobility. A major change was that the above-mentioned third region separates from the South Atlantic by two critical bridges (Fig 5c). One of the bridges shifted from within the Pennsylvania State to the border between New York and Pennsylvania State, dividing the two states. The other one emerged in Virginia State, separating the New England and Middle Atlantic from the Lynchburg Metropolitan Statistical Area in Virginia. The high infection cases likely caused

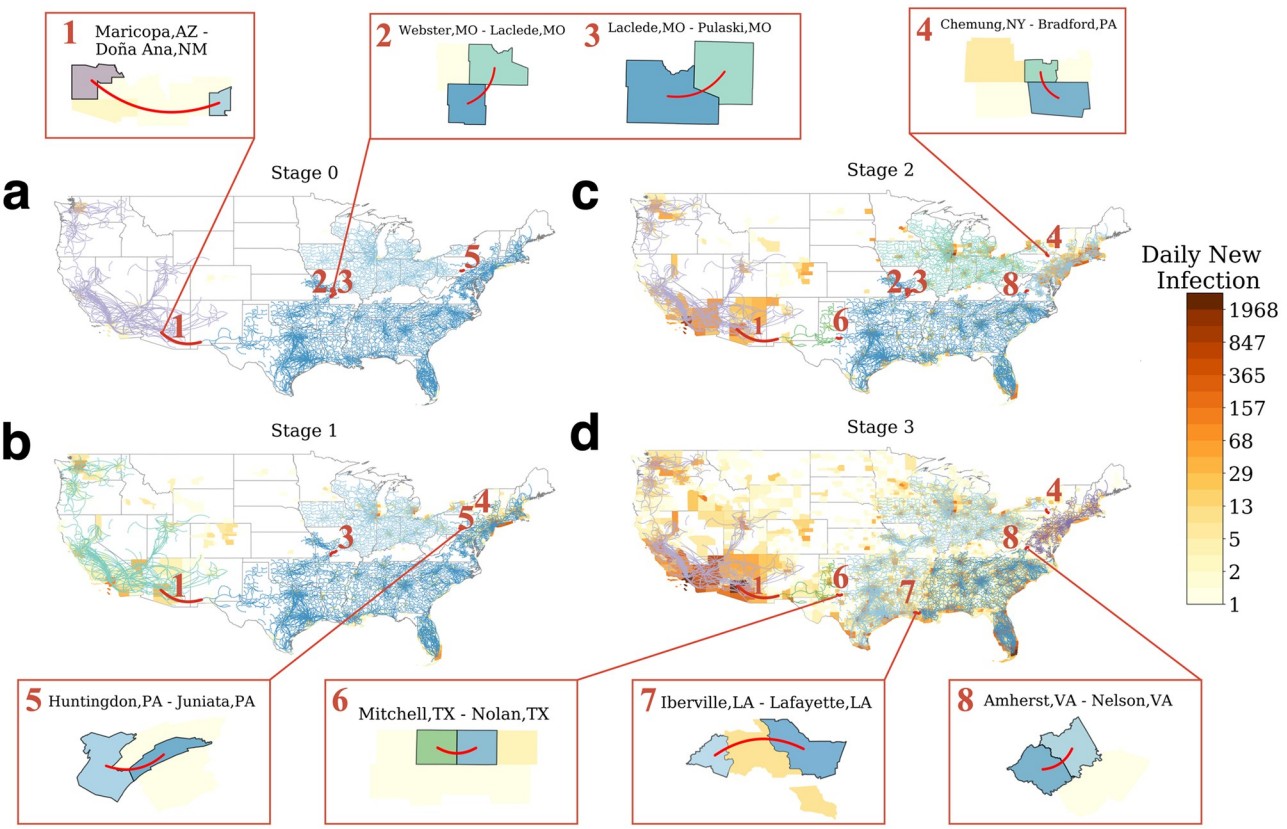

**Fig 5. Recurrent critical links detected at different stages. a, b, c, d**, components and recurrent critical links before the national emergency declaration (i.e., Stage 0), after the declaration from Mar. 13 to Mar. 27, 2020 (i.e., Stage 1), from Mar. 28 to Apr. 10, 2020 (i.e., Stage 2), and from Apr. 11 to Apr. 25, 2020 (i.e., Stage 3), respectively. The recurrent critical bridges in various periods are highlighted in red. These links are the edges of which the weights are just below the threshold of $q_c$ or $q_{c2}$. The removal of the edge between two nodes will disintegrate the functional components. The critical links were located near the borderlines between various sub-components. The heat map shows the average daily new infection case per county during the period on logarithmic scales.

the change in the epicenters (e.g., NYC, Boston, etc.) in the coastal areas at the early stage of COVID-19, which forced multiple levels of travel restrictions. Another critical link emerged in Texas, which further separates the West Pacific and the West South Central regions.

The recurrent critical links remained largely the same in the New England, Mid-Atlantic, and West Pacific divisions during the reopening stage (Stage 3, see Fig 5d) northeast mega-region remaining as separated clusters. However, there were some structural changes in the clusters. The most notable change is a new link connecting Iberville, LA, and Lafayette, LA, separating the East South Central and the East South Atlantic divisions. Also, the link in Missouri observed in Fig 5c reduced its criticality and connected the West South Central region to the Midwest region. The change is likely to be caused by the shifting of hot spots of infections during the reopening stage. At this time, the numbers of infection cases surged in the early reopening southern states such as Louisiana, Georgia, and Florida while remained stable or decreased in the Northeast and Mid-West regions. The previously identified links connecting both California and Texas could also increase to COVID-19 infections. The East Coast region's critical links remain largely attributed to the fact that early hot spots, e.g., New York and Boston, took cautious steps and gave strict guidance to reopen.

Overall, these results suggest that there are only a small number of recurrent critical bridges at each stage of the pandemic despite the unprecedented changes in mobility networks. They emerge due to the heterogeneity in both infection rate and mobility response in large clusters.

## The topological features to determine the critical transitions and adaptive bridges

An important question can be raised regarding the recurrent critical bridges: *are they the results of the structures of human mobility networks or simply a byproduct of artificial processes of any random networks?* To answer the question, we explore the topological features that determine the phase transitions and critical bridges in response to the propagation of COVID-19. We randomize the original networks and simulate their percolation processes. Then we adopted the same process as discussed before to identify new critical bridges in the randomized networks. Comparing the new bridges to the old ones allows us to validate that the critical bridges emerge from the network structures and the mobility perturbation caused by COVID-19 rather than random mechanisms [56]. Three randomization mechanisms were used to compare to the original network (Fig 6a): (1) randomly shuffling weights on original links (Fig 6b); (2) randomly rewiring the destinations of directed links (Fig 6c); and (3) randomly rewiring the origins of directed links (Fig 6d). The results show that the identified recurrent critical

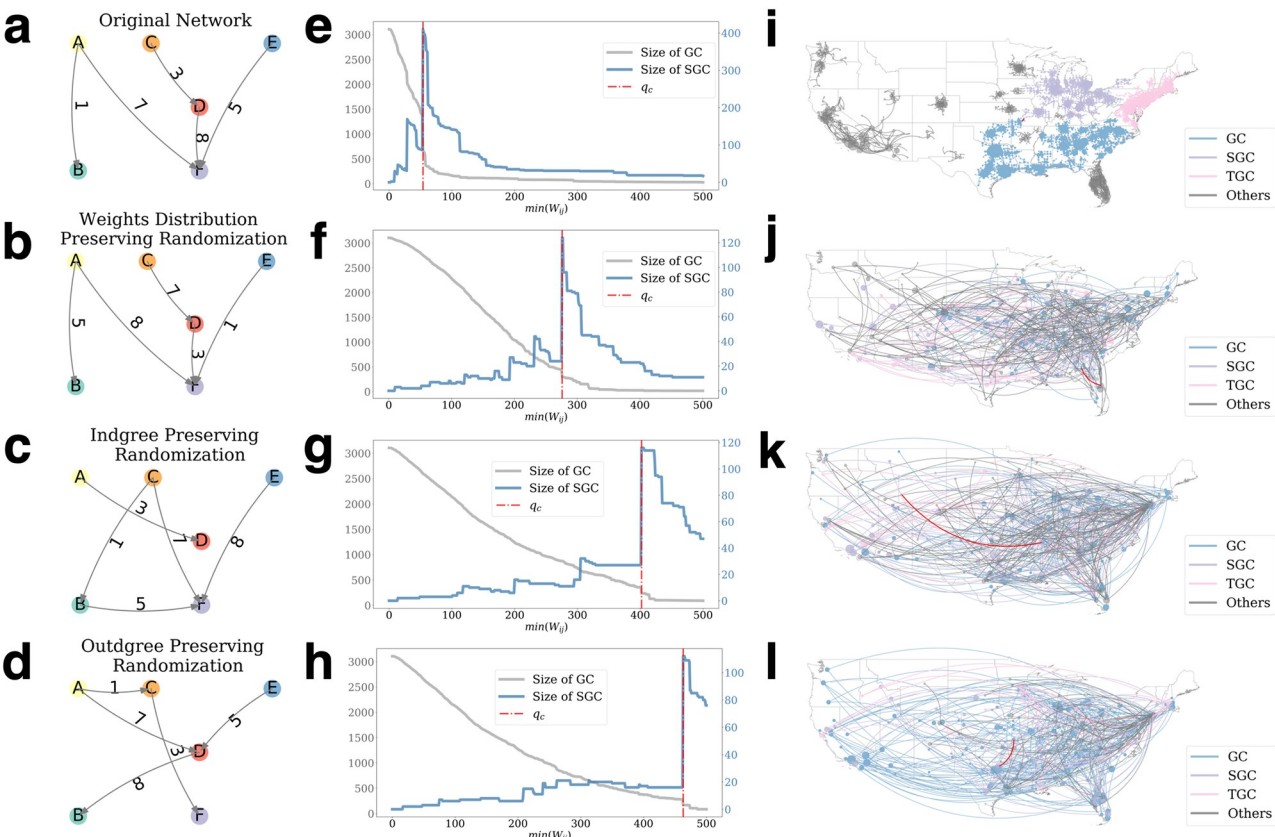

**Fig 6. Results from randomization and their comparison to the original mobility data. a** A sub-network of the original mobility network with their weights illustrated. The sub-network's percolation process is shown in **e** and its largest components at $q_c$ are shown in **i**. We applied three types of randomization to the original networks: randomly assigning weights with distribution unchanged (**b**), randomly shuffling the weights while each node's in-degree remain unchanged (**c**), and randomly shuffle the weights while each node's out-degree remain unchanged (**d**). Examples of the percolation processes of the three types are shown in **f, g, and h** respectively, and their largest components at $q_c$ are shown in **j, k, and l**.

links after randomization are completely inconsistent with the ones from original networks, indicating that the emergence of the critical links is not coincidental.

The results show that changing the weight alone without altering the structure would increase the $q_c$ from 107 to greater than 400, varying slighted in different randomizations (Fig 6). The change means it will take more than three times of the population to fragment the mobility networks into subcomponents and thus limit and control mobility in the U.S. Altering the outdegrees and indegrees would cause similar changes to $q_c$. Moreover, the phase transition observed in Fig 6a would become continuous changes in Fig 6c and Fig 6d. A phase transition means a small sacrifice of mobility can prevent a large proportion of the network from connecting; indeed, Fig 6a shows that by increasing our mobility threshold from 57 to 58, we can reduce the size of the largest connected component from 1436 to 731. Such effective control strategies would not be possible in continuous changes.

We furthermore randomize the daily network 1,000 times and observe how the randomized metrics shift as the daily mobility patterns change. Fig 7a refers to the shuffling demonstrated in Fig 6b where we can see that prior to the emergency state, the critical threshold remained lowest around 300. As the network perturbation takes place, the median of the critical threshold starts climbing correspondingly and declines as the mobility networks recover during the reopening stage. Also, Fig 7b captures the randomization result where the node in-degrees unaltered and witnesses similar trends with overall larger values of thresholds present. These results collectively suggest that in an arbitrary scenario, given the mobility perturbation, the critical thresholds would increase due to the absence of population and spatial propinquity effects. The increase suggests that in a random scenario, the abrupt changes of mobility networks could lead to a rise in the critical threshold. In this case, the perturbation increases the connectivity of mobility networks instead of reducing it, as we observed in real data. We compare Fig 4a and Fig 7 and find that COVID-19 changed the structure of the inter-county mobility network so that the $q_c$ decreases after national emergency day, indicating that the networks become easier to breakdown. In contrast, the corresponding randomized networks become more robust (larger $q_c$) to link removal after the national emergency day.

## Discussion

Our study aggregates the mobility network on the county level and does not consider movement within counties. Hence, the identified links of this study play a critical role in human travels, and possibly, the spread of COVID-19 between counties. Future research will focus on the geographies with higher resolution and examine the effect of intra-county mobility. Also, despite the unprecedented quantity and overall representativeness of our data sets (see S1 and S2 Figs in S1 File), the data could have selection bias originated from sociodemographic characteristics of the users, device types and functions, and usage behaviors. The high correlation ($r > 0.93$) through the six months demonstrates that the data are instrumental on the county level. Our results are valid for the large population of over 30 million users in this data set at a minimum.

Our results suggest that the mobility network experienced substantial perturbations, evidenced by the great and unusual changes in network features. Several metrics have decreased by over 50% even though the total population in our data only experienced a small decline. Our analytic approach from percolation theory allows us to capture universal mobility patterns even during perturbations. We reveal that the percolation criticality effect remains during the courses of COVID-19 spread. At each stage of the COVID-19, the overall network's critical state is comprised of large components connected by recurrent critical links. The findings

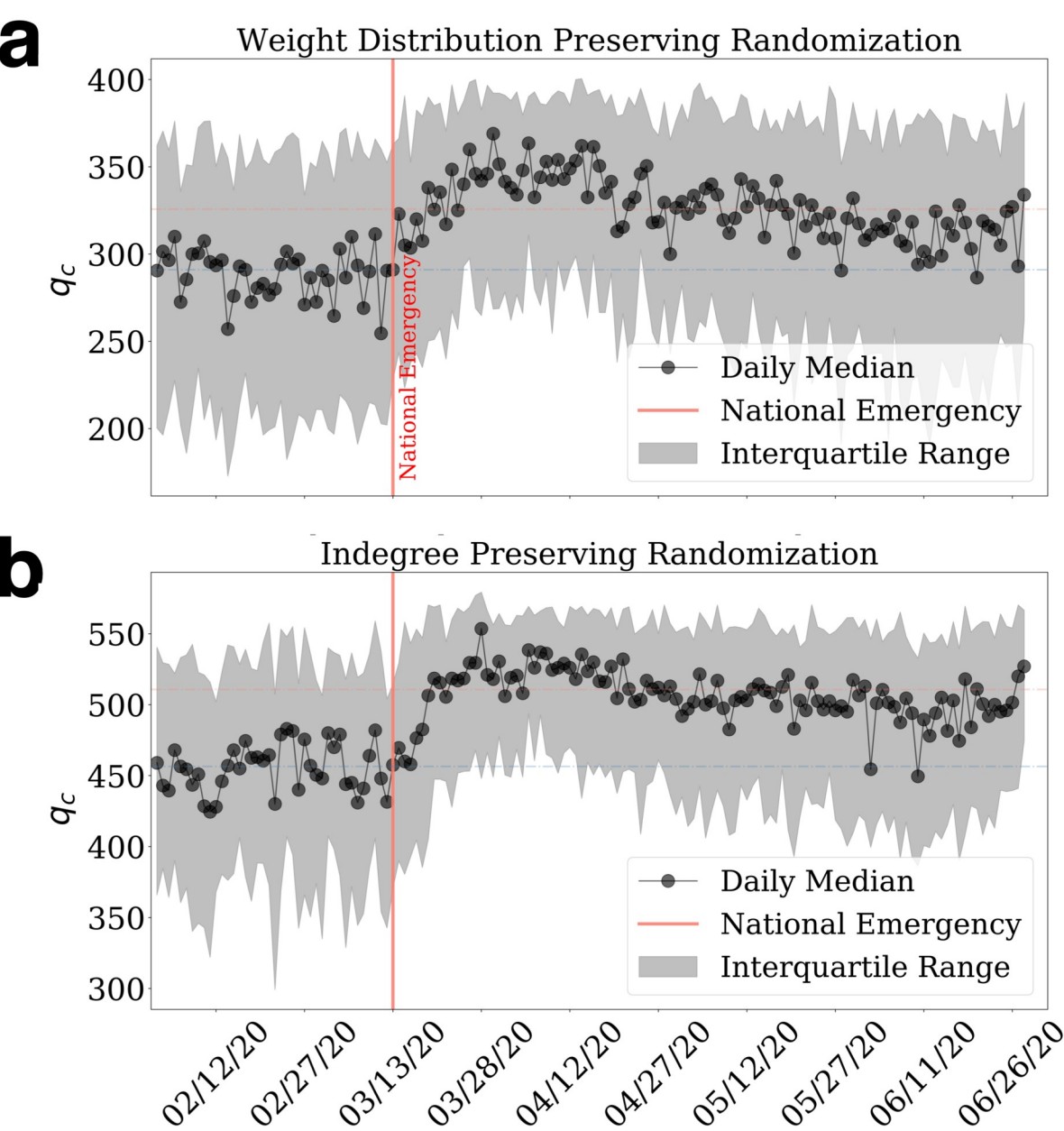

**Fig 7. Temporal changes of randomized percolation metrics. a**, the distribution of daily $q_c$ from 1,000 iterations of randomization that preserve network's weights. **b**, the distribution of daily $q_c$ from 1,000 iterations of randomization that preserve the indegrees of the network. The black dots represent the daily median value and the shaded areas indicate the interquartile range (IQR).

suggest that the established structures dominate the general patterns of the topology of mobility networks despite the heterogeneity of travel constraints between nodes.

The study uncovers some fundamental properties of human mobility when experiencing significant perturbations. For the first time, we show that the inter-county mobility networks exhibit surprisingly novel abrupt phase transitions. The structure of the network dominates the behaviors of the networks when disrupted. Particularly, mobility networks became vulnerable at the early stage of COVID-19, making the system more likely to reach the critical threshold. The drop in travels between counties after the declaration of national emergency has

reduced the strength of connections between nodes. The significant decrease of $q_c$ means the network reaches its criticality more easily and is more prone to fragmentation.

The complex behavior of mobility networks prompts our search and identification for a small, manageable set of recurrent critical bridges. Our analysis also shows that the emergence and perseverance of the critical links are non-trivial, meaning they do not appear due to randomness but are the product of the spatiotemporal patterns of human mobility. We confirm their existence and importance by showing their recurrence in different stages (Fig 5). These key links, located across the United States, played a key role as valves connecting components in divisions and regions. Also, some bridges were more persistent than others during the six months. The presence and lasting period correlate with mobility changes responding to stay-at-home orders and social distancing practices. Lastly, we found that the recurrent critical bridges do not appear randomly (Fig 5). Instead, they are mostly located at the edge of the components with relatively low populations. The finding is somewhat counter-intuitive and provides new insights into managing and controlling the connectivity of mobility-based networks.

Although the focus of this study is mobility network perturbation caused by COVID-19, the geographical distributions of the critical links highlighted their potential crucial roles in managing human mobility and consequently control the spread of COVID-19 [57]. We observe that the new emerging critical bridges appear to be at the border of the regions and divisions designated by the Census Bureau and close to the components within which the COVID-19 hot spots are located. Early identification, control, and even disconnections of these recurrent critical links these links could dissolve the mobility networks into small components and contain physical contacts within sub-components. Practices and policies focusing on these links can limit disease diffusion locally and thus delay and even prevent cross-component infections.

There are still some limitations in this study fostering some directions for further studies. First, the aggregated GPS data may not include individuals without smartphones and could overestimate those who have multiple devices. Moreover, our study focuses on the inter-county mobility network only. The relation between local mobility dynamics and the spread of COVID-19 also merits scrutiny and can be analyzed by more comprehensive and detailed modeling of people's movements locally. Such as map-matching trajectory to POIs with modeling of temporal visitation patterns. Finally, the aggregation and compression of mobility data presented in this study limits to only one county. An opportunity for future work is to formulate methods that can capture the detailed spatiotemporal movements of each user.

## Conclusion

Human activities, especially mobility-enabled social interactions, have become the "key determinant of disease emergence" [58]. Our percolation-based approaches allow the identification of critical thresholds for mobility networks. Thus, it provides new knowledge of the complex networks under the influence of external disruptions. The method can also be utilized to develop tools to forecast phase transition, i.e., both collapse and recovery, of mobility networks. Also, we show it is a powerful tool for fast and accurate identification of critical bridges in highly dynamic mobility networks. These bridges help predict critical transmission paths on different scales. They are useful in managing and control mobility during large-scale perturbations caused by other types of natural disasters and extreme events.

## Supporting information

**S1 File.**
(PDF)

## Author Contributions

**Conceptualization:** Jing Du, Jianxi Gao, Qi Wang.

**Data curation:** Hengfang Deng, Qi Wang.

**Formal analysis:** Hengfang Deng.

**Funding acquisition:** Jing Du, Jianxi Gao, Qi Wang.

**Investigation:** Hengfang Deng.

**Methodology:** Hengfang Deng, Qi Wang.

**Supervision:** Jianxi Gao, Qi Wang.

**Validation:** Hengfang Deng.

**Visualization:** Hengfang Deng.

**Writing – original draft:** Hengfang Deng.

**Writing – review & editing:** Hengfang Deng, Jing Du, Jianxi Gao, Qi Wang.

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
