## [Decision Letter · Decision Letter 0]

15 Jul 2021

PONE-D-21-17570

Network percolation reveals adaptive bridges of the mobility network response to COVID-19

PLOS ONE

Dear Dr. Wang,

Thank you for submitting your manuscript to PLOS ONE. After careful consideration, we feel that it has merit but does not fully meet PLOS ONE’s publication criteria as it currently stands. Therefore, we invite you to submit a revised version of the manuscript that addresses the points raised during the review process.

Both reviewers made positive comments about the manuscript and the study it describes. However they also made some criticisms and suggested some improvements. I would therefore encourage you to submit a revised version of your manuscript, taking into account the clarifications and methodological questions raised by Reviewer 1, as well as the corrections suggested by both reviewers.

We look forward to receiving your revised manuscript.

Kind regards,

Gareth J. Baxter

Academic Editor

PLOS ONE

Journal Requirements:

3. We note that Figures 1-5, S4, S6 in your submission contain map images which may be copyrighted. All PLOS content is published under the Creative Commons Attribution License (CC BY 4.0), which means that the manuscript, images, and Supporting Information files will be freely available online, and any third party is permitted to access, download, copy, distribute, and use these materials in any way, even commercially, with proper attribution. For these reasons, we cannot publish previously copyrighted maps or satellite images created using proprietary data, such as Google software (Google Maps, Street View, and Earth). For more information, see our copyright guidelines: http://journals.plos.org/plosone/s/licenses-and-copyright.

3.1.    You may seek permission from the original copyright holder of Figure 1-5,  S4, S6 to publish the content specifically under the CC BY 4.0 license. 

3.2.    If you are unable to obtain permission from the original copyright holder to publish these figures under the CC BY 4.0 license or if the copyright holder’s requirements are incompatible with the CC BY 4.0 license, please either i) remove the figure or ii) supply a replacement figure that complies with the CC BY 4.0 license. Please check copyright information on all replacement figures and update the figure caption with source information. If applicable, please specify in the figure caption text when a figure is similar but not identical to the original image and is therefore for illustrative purposes only.

Reviewers' comments:

Reviewer's Responses to Questions

**Comments to the Author**

1. Is the manuscript technically sound, and do the data support the conclusions?

Reviewer #1: Yes

Reviewer #2: Yes

2. Has the statistical analysis been performed appropriately and rigorously? 

Reviewer #1: Yes

Reviewer #2: I Don't Know

3. Have the authors made all data underlying the findings in their manuscript fully available?

Reviewer #1: No

Reviewer #2: Yes

4. Is the manuscript presented in an intelligible fashion and written in standard English?

Reviewer #1: Yes

Reviewer #2: Yes

5. Review Comments to the Author

Reviewer #1: This paper uses mobility traces collected from volunteers to detect differences in human mobility patterns between time periods immediately preceding and after the introduction of stay-at-home-orders due to the COVID-19 pandemic. The analysis performed by the authors focuses on graphs constructed by the amount of inter-county travel in the United States and the partitioning thereof. The authors find that the introduction of stay-at-home orders caused short term disruptions in the mobility network and that critical links to graph partitioning could be determined.

Generally, the topic is resonates with the times. It is clearly an important topic and will be of general interest to readers. The findings, while not altogether surprising provide an interesting framework for discussion of the role of human mobility in the spread of disease and how network analysis can be used to track diseases and possibly influence policy to better control the spread.

The idea of using the framework of phase transitions and finding q_c is an interesting one and the numerical analysis provided is solid. Similarly, the randomization steps taken to ensure that these are a product of human behavior and not merely products of a random network lend credibility to the work.

However, there are things in the paper that are poorly explained that could stand better explanation.

1. It is not clear what the "median location" of a person is or how frequently these locations are computed. The paper rests on this idea, and yet it is not fully explained. If these are continuous GPS traces, is it the location at which the person is at noon? Are the GPS locations taken every 15 minutes and this is in fact the "average" location?

2. Similarly, percolation and phase transition theory come from physics, which is not the core audience of this paper, a brief introduction to what this is and what to look out for/why it is appropriate would help the reader immensely.

3. Eyeballing Fig.2 and Fig 4. it doesn't look like the timing of the upswing is particularly different but the authors make the assertion on page 4/10 that these are significantly different. A stronger statement in that direction or justification would be nice.

4. Similarly, the line in Fig. 4 for "mean value - after" is clearly only the mean value for some time-window after, but it is not clear what that window is.

5. The significance of the "critical links" discussed on page 5 is somewhat dubious to me. It seems that the weighting on a link does not change drastically from time period to time period, and thus it's location in an ordered list of links weights would not shift drastically. Thus, a link that causes the "important" partition once is relatively likely to be in the same place again. I am not sure how the identification of this fact helps, since the remove of links in the network is a numerical not factual exercise.

6. County size might vary dramatically across the united states. While restricting a user to one county per day might work in some places, I would assume that there are places in the united states for which users travel in multiple counties in a given day regularly. The "one county per day" restriction seems unnecessarily limiting. How would things have changed if the time granularity were an hour? I suppose this is related to Page 2: "We average the networks from 7 days" but it is not clear what that means either, how does that averaging happen?

Smaller issues that need addressing:

1. On page four, in the middle: "Despite the network metrics steadily recovered" has grammatical issues, as does the rest of that paragraph.

2. Final paragraph of page 4/10 has at missing \\ in the LaTeX (bad section heading)

3. The first and last paragraphs of page 6/10 seem to be the same paragraph

4. In the abstract, I don't understand "The mobility network becomes vulnerable and prone to reach its criticality"

Reviewer #2: The authors analyze the large-scale human mobility networks covering the first 6 months during the pandemic of COVID-19. They show, regardless of inherent complexity of human mobility, that these networks clearly exhibit abrupt phase transitions, in contrast with the understanding of traditional percolation models where a continuous phase transition is expected. They develop new approaches to identify a set of recurrent critical links, connecting the giant component and the second-largest component, and show that these links are associated with network characteristics. This work shows a clear practical relevance and a general applicability in analyzing human mobility behaviors under interruptions of different kinds of infectious diseases. In addition, the paper is clearly written and easy to follow. It is a very good paper. Before recommending for publication, few minor comments are listed as follows.

1 In the human mobility data, besides the two critical weight q_c and q_c2, do you observe more critical weights? This would for sure depend on a couple of factors like the total time span of the available data, the jump size and time window. I am wondering, within the available data, are there other critical weights in different parameter settings?

2 The results shown in Fig 2 indeed align with intuitive understanding that after the release of stay-at-home-orders, human mobility would decline. I am wondering are there variability and dependence with other factors like selected time window, either right after the declaration or even later than the studied one. In addition, are there some relevance with the daily infections? For instance, an increased daily infection would reinforce the emergency of current situation and mobility might accordingly drops.

3 At the current presentation, the resolution of Fig 1 is lower than other figures. It would be easier to follow with a higher resolution.

4 At page 3, Fig3b and Fig3c mismatch the text description. Last paragraph in page 4 is intended to be highlighted? At page 5, network structure instead of stricture. At page 2, please check Fig. 2b one.

6. PLOS authors have the option to publish the peer review history of their article (what does this mean?). If published, this will include your full peer review and any attached files.

Reviewer #1: No

Reviewer #2: No

---

## [Author Response · Author response to Decision Letter 0]

6 Sep 2021

Pleases see the attached "Response to Reviewers" file.

---

## [Decision Letter · Decision Letter 1]

30 Sep 2021

PONE-D-21-17570R1Network percolation reveals adaptive bridges of the mobility network response to COVID-19PLOS ONE

Dear Dr. Wang,

Thank you for submitting your manuscript to PLOS ONE. After careful consideration, we feel that it has merit but does not fully meet PLOS ONE’s publication criteria as it currently stands. Therefore, we invite you to submit a revised version of the manuscript that addresses the points raised during the review process. Both reviewers are happy with the revised manuscript. However each indicated one small further adjustment that they would like to see made. I have therefore made the decision "minor revision" merely to allow you the chance to make these revisions.

We look forward to receiving your revised manuscript.

Kind regards,

Gareth J. Baxter

Academic Editor

PLOS ONE

Journal Requirements:

Reviewers' comments:

Reviewer's Responses to Questions

**Comments to the Author**

1. If the authors have adequately addressed your comments raised in a previous round of review and you feel that this manuscript is now acceptable for publication, you may indicate that here to bypass the “Comments to the Author” section, enter your conflict of interest statement in the “Confidential to Editor” section, and submit your "Accept" recommendation.

Reviewer #1: (No Response)

Reviewer #2: (No Response)

2. Is the manuscript technically sound, and do the data support the conclusions?

Reviewer #1: Yes

Reviewer #2: Yes

3. Has the statistical analysis been performed appropriately and rigorously? 

Reviewer #1: Yes

Reviewer #2: Yes

4. Have the authors made all data underlying the findings in their manuscript fully available?

Reviewer #1: Yes

Reviewer #2: Yes

5. Is the manuscript presented in an intelligible fashion and written in standard English?

Reviewer #1: Yes

Reviewer #2: Yes

6. Review Comments to the Author

Reviewer #1: This paper continues to be an interesting paper and I look forward to seeing it published. The authors have addressed all my previous concerns but one that I would like to see addressed. In the authors response to my previous review, they explained that they calculated "median location" as the median of the latitude and longitude. It is easy to see that this may result in a location the user has not visited at all or spends very little time in (my own median location is likely somewhere on a highway in a county between my home county and work county). Nevertheless, I understand that the method is borne of necessity given the scale of the dataset and do not take over much issue with it. What I would like to see is some text in the published version of the paper that explains that this is how they have calculated location for a user. The current discussion is a single sentence that reads "We find the median of the locations from each user’s daily movement trajectory and compute the primarily located county on each day(see Fig 1a)." which I feel is insufficient to explain what they have done. The paragraph they included in the response to reviewers is great and compelling, but I would be satisfied with a single parenthetical: "(i.e., the location of the medians of longitude and latitude)." It is a truly minor change. However, as the whole paper rests on where a user is considered to be, a strong understanding of how it is derived is crucial.

Reviewer #2: The authors’ response and modifications in the MS have clarified nearly all the concerns. A remaining issue is that after the replacement of fig1, it is still difficult to clearly read info in fig1, for example user labels in fig1b. After this issue has been addressed, I opt for a recommendation for publication.

7. PLOS authors have the option to publish the peer review history of their article (what does this mean?). If published, this will include your full peer review and any attached files.

Reviewer #1: No

Reviewer #2: No

---

## [Author Response · Author response to Decision Letter 1]

4 Oct 2021

Please see our "response to reviewer" file.

---

## [Editor Report · Decision Letter 2]

7 Oct 2021

Network percolation reveals adaptive bridges of the mobility network response to COVID-19

PONE-D-21-17570R2

Dear Dr. Wang,

We’re pleased to inform you that your manuscript has been judged scientifically suitable for publication and will be formally accepted for publication once it meets all outstanding technical requirements.

Kind regards,

Gareth J. Baxter

Academic Editor

PLOS ONE
---

## [Editor Report · Acceptance letter]

19 Oct 2021

PONE-D-21-17570R2 

Network percolation reveals adaptive bridges of the mobility network response to COVID-19 

Dear Dr. Wang:

I'm pleased to inform you that your manuscript has been deemed suitable for publication in PLOS ONE. Congratulations! Your manuscript is now with our production department. 

Kind regards, 

on behalf of

Dr. Gareth J. Baxter 

Academic Editor

PLOS ONE